# Gender and Allergy: Mechanisms, Clinical Phenotypes, and Therapeutic Response—A Position Paper from the Società Italiana di Allergologia, Asma ed Immunologia Clinica (SIAAIC)

**DOI:** 10.3390/ijms26199605

**Published:** 2025-10-01

**Authors:** Maria Teresa Ventura, Antonio Francesco Maria Giuliano, Elisa Boni, Luisa Brussino, Rosalba Buquicchio, Mariaelisabetta Conte, Maria Teresa Costantino, Maria Angiola Crivellaro, Irene Maria Rita Giuliani, Francesca Losa, Stefania Nicola, Paola Parronchi, Silvia Peveri, Erminia Ridolo, Paola Triggianese, Vincenzo Patella

**Affiliations:** 1Medical School, University of Bari “Aldo Moro”, 70121 Bari, Italy; mitiventura@gmail.com; 2Clinica Medica “A. Murri”, Department of Preventive and Regenerative and Ionian Area (DiMePre-J), University of Bari “Aldo Moro”, AOUC Policlinico Hospital, 70124 Bari, Italy; 3UOS Immunologia Clinica, Ospedale Maggiore, 40133 Bologna, Italy; 4Department of Medical Sciences, University of Turin, 10128 Turin, Italy; stefania.nicola@unito.it; 5Immunology and Allergy Unit, AO Ordine Mauriziano di Torino, 10128 Turin, Italy; 6Dermato-Oncology Unit, IRCCS Istituto Tumori “Giovanni Paolo II”, 70124 Bari, Italy; r.buquicchio@oncologico.bari.it; 7Struttura Complessa di Allergologia e Immunologia di Laboratorio, Azienda Sanitaria Friuli Occidentale (ASFO), Ospedale S. Maria degli Angeli, 33170 Pordenone, Italy; mariaelisabetta.conte@asfo.sanita.fvg.it; 8Struttura Complessa di Allergologia, Immunologia Clinica e Reumatologia, ASST Mantova, 46100 Mantova, Italy; mariateresa.costantino@asst-mantova.it (M.T.C.);; 9Departmental Allergy Division, Department of Systems Medicine, University Hospital of Padua, 35128 Padua, Italy; mariaangiola.crivellaro@unipd.it; 10Allergology and Clinical Immunology Unit, Department of Medicine and Surgery, University of Parma, 43126 Parma, Italyerminia.ridolo@unipr.it (E.R.); 11Department of Experimental and Clinical Medicine, University of Florence, 50134 Florence, Italy; paola.parronchi@unifi.it; 12Immunology and Cell Therapy Unit, University Hospital Careggi, 50134 Florence, Italy; 13UOSD Allergologia, AUSL Piacenza, 29121 Piacenza, Italy; s.peveri@ausl.pc.it; 14Department of Biomedicine and Prevention, University of Rome Tor Vergata, 00133 Rome, Italy; paola.triggianese@gmail.com; 15Department of Internal Medicine ASL Salerno, ‘Santa Maria della Speranza’ Hospital, 84131 Salerno, Italy; patella@allergiasalerno3.it; 16Postgraduate Program in Allergy and Clinical Immunology, University of Naples ‘Federico II’, 80131 Naples, Italy

**Keywords:** position paper, SIAAIC, allergy, gender, phenotypes, immune responses

## Abstract

Sex and gender play a critical role in allergic diseases, influencing immune response, clinical phenotypes, treatment strategies, outcomes, and health-related quality of life. Despite mounting evidence across multiple studies examining sex/gender differences in a multitude of allergic diseases, most address isolated conditions, not taking into consideration the vast interplay of hormonal, genetic, immunological, and sociocultural factors and their unique consequences for clinicians and researchers. With this position paper, we aim to assess currently available evidence on the sex- and gender-specific characteristics of the most common allergic diseases, providing an overview of present knowledge and future areas of improvement for clinicians and researchers. This position paper was developed by the *Società Italiana di Allergologia, Asma ed Immunologia Clinica (SIAAIC)*: a panel of experts who conducted a literature review focusing on sex and gender differences across major allergic diseases. A consensus-based approach was employed to assess the immunological, clinical, and therapeutic implications of available evidence, offering a recommendation for researchers and clinicians alike. Data highlights marked differences driven by sex and gender in disease prevalence, immune pathways, clinical phenotype and severity, as well as therapeutic outcomes. Female patients appear to show a higher prevalence of Th2-driven ailments, autoimmune overlap, and allergic drug reactions, whereas males are more likely to experience fatal anaphylaxis and severe mastocytosis. Sex hormones can modulate multiple immune pathways leading to mast cell activation, antibody production, and cytokine expression, thus contributing to divergent disease trajectories. In conclusion, sex and gender are a key determinant in allergic diseases, and their integration in future research is essential to develop a tailored approach to treatment. Efforts should prioritise the identification of sex- and gender-specific biomarkers, therapeutic strategies, and equitable access to healthcare services. A sex- and gender-aware approach could potentially improve outcomes, optimise treatment strategies, and address current gaps in allergy practice.

## 1. Introduction

In recent years, medical science has moved beyond the concept of “one size fits all”, recognising how diseases can greatly vary in epidemiology, clinical phenotype, and treatment strategy according to each patient’s unique characteristics. Studies derived from real-world data have underlined the multitude of factors that directly influence the course of disease, often markedly differing from the results of clinical trials [1]. Two key differences are sex and gender: until the past two decades, many studies and trials did not fully explore potential differences between male and female patients, often excluding women altogether or operating under the assumption that their molecular and phenotypical characteristics were largely similar [2]. Biological sex, gender identity, and societal role enter a complex interplay with hormonal, immunological, and metabolic pathways that may lead to unique and often unmet healthcare needs [3]. Along with age, literacy, and environmental characteristics, these factors may lead to sizable repercussions on clinical outcomes and health-related quality of life [4]. Such an approach is especially critical in allergic diseases: sex is a key variable, influencing lymphocyte differentiation and function, overall immune response, clinical characteristics, treatment effectiveness, and outcomes, resulting in unique fluctuations in quality of life. Although multiple studies in the field have examined sex and gender differences in allergic diseases, most offer a highly specialised point of view, not taking into consideration the possible differences in aetiological pathways and the unique consequences for clinicians and researchers.

With this position paper, we aim to assess currently available evidence on sex- and gender-specific characteristics of the prevalent allergic diseases, providing an overview of present knowledge and future areas of improvement.

## 2. Methods

This position paper was commissioned by the Società Italiana di Allergologia, Asma ed Immunologia Clinica (SIAAIC), of which the Authors are members. The Society strongly maintains that the exploration of sex and gender differences in allergic diseases is essential to improve precision medicine and patient care. By supporting this work, SIAAIC aims to underscore the scientific and clinical relevance of sex- and gender-tailored strategies in allergy, asthma, and immunology, thereby fostering a more comprehensive approach. The Authors performed a literature search on MEDLINE and PubMed, combining search terms such as “sex differences”, “gender differences”, and “allergy” with nine allergic diseases, selected by panel members as the most clinically and socially relevant: allergic rhinitis, asthma, anaphylaxis, allergic drug reactions, mastocytosis, urticaria, atopic dermatitis, gastrointestinal allergies, and professional allergies. There were no restrictions on publication year, and search was limited to English-language papers. Titles and abstracts were screened for relevance, while non-human studies and isolated case reports were excluded. Each section was assigned to one or more Authors, who reviewed the relevant literature and prepared a first draft. Revisions were then circulated to the full group, discussed collectively, and revised until agreement was reached. Disagreements were resolved through discussion and majority opinion. The Delphi method was used throughout the process to ensure an unbiased consensus.

## 3. Immunological Aspects of Allergic Diseases

Immune homeostasis is maintained by control mechanisms and regulatory pathways in which hormonal axes play a key role. Sex hormones are the main mediators for differences in susceptibility to specific diseases, including immune-mediated and allergic ones, as well as the immune response to infections, vaccines, and malignancies [5,6,7]. Sex hormones exert their biological effects by binding to nuclear receptors, functioning as ligand-activated transcription factors. Upon hormone binding, these receptors undergo conformational changes, and, translocating into the nucleus, regulate the expression of target genes. Oestrogens, in particular 17β oestradiol (E2), interact with oestrogen receptor (ER) α and ERβ, expressed by all types of immune cells: depending on the specific receptor, effects can be both immunostimulatory (by ERα) or immunosuppressive (by ERβ). Inflammatory responses induced by type 2 innate lymphoid cells can be enhanced by progesterone, likely contributing to exacerbations of immune-mediated diseases [8].

Androgens exert an overall suppressive effect on immune cells by inhibiting T helper 1 differentiation, and thus immune responses mediated by T1 pro-inflammatory cytokines. Furthermore, evidence suggests a key role of androgens in regulating B cell development along with their effects in inhibiting autoreactive B cell expansion and thus reducing autoantibody production [9]. Experimental evidence and epidemiological research support the role of oestrogens in modulating the immune response and consequently their physiopathological role in immune-mediated diseases. It is conceivable that during pregnancy the evolutionary basis for “survival” and immune system adaptation mechanisms constitutes the most relevant sex-related difference. Through a correct balance between inflammation and anti-inflammatory mechanisms, the maternal immune system can tolerate the embryo and therefore the foetus until childbirth [10]. During pregnancy, three distinct immunologic phases can be characterised: a pro-inflammatory environment during embryo implantation, placentation, and the early stage of pregnancy; an anti-inflammatory milieu during mid-pregnancy; and a pro-inflammatory environment at the third trimester and at the end of pregnancy [10]. This increased immunoreactivity in the female sex makes women more resilient against infections and capable of a more powerful serological response; at the same time, it supports the increased susceptibility to autoimmune diseases. Moreover, the oestrogenic-mediated cellular hyperactivation, mast cell reactivity, and delayed type IV response contribute to the symptomatic profile in allergic diseases documented in female patients [8,11,12]. As for autoimmune diseases, allergies are closely associated with the sex-linked hormonal axis, determining distinct symptomatic profiles by modulating immune responses. Atopic conditions such as allergic asthma and atopic dermatitis are more prevalent in females than in males [8,13]. Age-related changes play a significant role in the distribution of the prevalence of autoimmune and allergic diseases, as in males the latter have a higher incidence until puberty, after which the prevalence and severity ratio is reversed [8,13]. The stage of evolutionary development is a further element to consider in immunological maturation since it contributes to changes in the hormonal curves and responses to sexual hormones. It should be noted that thyroid hormones present well-documented interactions with immune cells, possibly playing a role in the pathogenesis of allergic diseases, especially in females [14,15]. A similar role has been hypothesised for prolactin, whose receptors have been documented in cells of the innate and adaptive immune system [16], with hyperprolactinemia causing abnormalities in immune response [16,17]. Of note, sexually dimorphic mast cell responses were observed in murine models subjected to immunological or psychological stress. Compared with male mice, females exhibited increased serum histamine levels and had greater intestinal permeability. Primary bone marrow mast cells from female mice exhibited increased mediator release, associated with markedly increased capacity for synthesis and storage of MC granule-associated immune mediators [18].

*Recommendations:* Sex hormones exert significant direct and indirect effects on immune regulation, contributing to sex-specific disease profiles in allergic and autoimmune disorders. Clinicians should take these influences into consideration with a sex-aware approach to patient management. Research efforts should further investigate hormone–immune interactions to inform such tailored strategies.

## 4. Allergic Rhinitis

Allergic rhinitis (AR) is a widespread condition that significantly impacts the quality of life of millions globally, with its prevalence and severity varying significantly with sex, gender, and age. The literature data consistently shows a predominance of symptomatic AR in males during early childhood [19]. During adolescence and adulthood however, females are more likely to develop AR. The transition typically occurs during puberty, [20] as reported by a meta-analysis by Pinart et al. [19]. According to a recent analysis conducted on an American cohort, this change in prevalence remains stable in adulthood: adult females are more affected by AR than males, with 23% increased odds of suffering from allergic rhinitis, while the opposite is true in the elderly [19,21]. The reasons for this shift remain poorly understood, although hormonal changes associated with puberty are likely a significant contributing factor: oestrogen and progesterone increase susceptibility to allergic conditions in females, while androgens may suppress immune responses in males, delaying the onset of symptoms [22]. This hormonal influence may also impact the severity of symptoms, with females potentially experiencing more severe reactions to allergens due to the inflammatory effects of oestrogens on the immune system [23]. Geographic and cultural variations are further variables influencing the sex-based prevalence of AR. In some regions, particularly in parts of Asia, male predominance in AR persists into adolescence and adulthood, as seen in studies from Japan, China, and South Korea [19]. These regional differences suggest that environmental exposures, genetic predispositions, and healthcare access disparities may also contribute to different AR prevalence between sexes. Sex differences may influence response to treatment, with a recent Chinese study suggesting better short-term efficacy of subcutaneous immunotherapy (SCIT) in females. Males are less adherent due to poorer early results, although the efficacy tends to converge between sexes over longer times [24]. Treatment during pregnancy, usually related to the worsening of AR and the onset of rhinitis gravidarum, should be carefully considered and personalised. Allergen immunotherapy (AIT) can continue if well-tolerated, but initiating treatment during pregnancy is not recommended [25]. Oral corticosteroids and decongestants should be avoided, particularly during the first trimester, due to potential teratogenic effects and associations with birth defects [26]. Pregnancy rhinitis, affecting 20–30% of pregnant women and potentially occurring alongside AR, should preferably be treated with non-pharmacological interventions, such as saline nasal rinses. Hyaluronic acid-based nasal washes may provide additional relief [27]. Topical steroids should not be prescribed for this kind of rhinitis for lack of effects, despite their safety in pregnancy.

*Recommendations:* There is an urgent need to develop sex- and gender-specific treatment strategies. Early diagnosis of AR should be pursued to achieve the best management according to gender and age influences [28]. As such, early intervention in males may prevent the development of more severe symptoms later in life, whereas females may benefit from a treatment approach accounting for hormonal fluctuations. Furthermore, cultural and environmental factors need to be considered when contemplating treatment strategies, owing to their role in the prevalence of AR.

## 5. Asthma

Asthma remains a major public health concern, affecting millions worldwide [29]. There are distinct patterns of incidence, prevalence, and severity across the lifespan of an individual, with notable differences between males and females. In early childhood, asthma is significantly more prevalent in boys (11.9%, vs. 7.5%) [30], with males twice as likely to be hospitalised for exacerbation [31,32]. Symptoms often improve with age, leading to a significant decline in incidence during adolescence [33]. As individuals mature, male predominance in asthma burden diminishes and is eventually reversed by puberty. Early menarche has been associated with an increased risk of developing asthma [34] and approximately 30–40% of afflicted women report premenstrual worsening of asthmatic symptoms. During adulthood, women consistently demonstrate a greater asthma prevalence (9.6%, vs. 6.3%) [30] and are three times more likely to be hospitalised for an asthma-related event [33,35]. These disparities narrow during the fifth decade, with a secondary increase in asthma incidence among middle aged men [30,36,37]. Occupational exposures, smoking history, and age-related changes in immune function and airway remodelling may influence this trend. Additionally, menopause represents a significant physiological hormonal transition whose impact on asthma is still under debate. Some studies suggest, in fact, that asthma prevalence decreases in postmenopausal women [38], with some research indicating a worsening in symptoms during the menopausal transition [39,40]. Sex hormones are the key actors in gender-specific differences. Oestrogens have been shown to enhance Th2-mediated airway inflammation, increasing eosinophilic infiltration, mucus production, and airway hyperreactivity [41,42]. Oestrogen can, in fact, modulate airway inflammation through multiple pathways, including its effects on innate lymphoid cells (ILCs) which play a crucial role in type-2 inflammation. These cells are more active in females than in males, contributing to increased asthma severity in women [43]. Additionally, oestrogen enhances IL-17A-mediated airway inflammation, which is more pronounced in females with severe asthma compared to males [44] and correlates with neutrophilic inflammation [45]. Moreover, neutrophilic inflammation may represent a pivotal pathogenic mechanism underlying asthma in obese women, in parallel with the involvement of the mTOR signalling pathway, as highlighted by several authors [46,47]. On the other hand, progesterone’s impact on asthma is more complex, as it can have both pro- and anti-inflammatory effects [41,43]. The fluctuations in progesterone levels during the menstrual cycle correlate with worsening asthma symptoms and the need for higher doses of corticosteroids [48,49] in some women, a phenomenon termed perimenstrual asthma (PMA) [48]. In contrast to oestrogen and progesterone, testosterone appears to exert a protective effect against asthma, as numerous studies showed that it can reduce Th2-driven inflammation, inhibit eosinophilic activity, and modulate airway remodelling [50], further confirming its regulatory role [51]. Beyond hormonal influences, genetic, epigenetic, and post-transcriptional regulatory mechanisms contribute to the distinct sex differences observed in asthma: in particular, the X chromosome is linked to unique immune profiles, as certain genes escape its inactivation and are expressed at higher levels compared to males [52]. Molecular mechanisms can not only influence asthma susceptibility but also impact treatment outcomes, particularly regarding corticosteroid responsiveness, which tends to be lower in women with severe asthma.

*Recommendations:* Hormonal changes, genetic predisposition, and environmental factors underscore key differences in asthma epidemiology, reinforcing the need for sex-specific approaches in management and treatment. Sex hormones regulate immune responses, airway inflammation, and bronchial hyperresponsiveness, influencing the onset and severity of asthma at different life stages. A greater understanding of these sex-specific differences is crucial for developing personalised treatment approaches. Future research should focus on tailoring interventions to address the unique immunological and hormonal factors that influence asthma pathophysiology in men and women. By integrating sex-specific considerations into clinical practice, it may be possible to optimise treatment efficacy and improve long-term outcomes.

## 6. Anaphylaxis

Anaphylaxis is a life-threatening, rapid onset, generalised or systemic hypersensitivity reaction involving at least two different organs across the skin, respiratory, cardiovascular, or gastrointestinal systems. Analogously to other allergic diseases, anaphylaxis is more frequent in females rather than in males, with the former accounting for more than half of the cases independently of symptoms. Available data is conflicting: [53,54] a UK study suggests that women are affected by more severe symptoms (IR 2.55 vs. 2.39), especially when suffering from severe asthma (IR 19.57 vs. 7.56) [55]. Vice versa, a higher incidence in males was found in Asian countries, such as Korea and Taiwan [56,57]. The main sources of bias in these population studies are data inhomogeneity, whether derived from real-world data or electronic health records, and the lack of stratification by the causative agent. Sex- and gender-specific differences in exposure to triggers are undoubtedly relevant, for example, in occupational anaphylaxis [58]. Similarly, evidence in food allergy-related anaphylaxis is ambiguous [59]. According to data from the FAERS database from 1999 to 2019, 62.71% of patients reporting anaphylaxis from medications were female, with antibiotics being the most frequent cause [60]; similar results were found in an American cohort [61]. These findings contradict a recent Japanese study, where sex did not appear to be an influencing factor, although fatalities were higher among males [62]. Idiopathic anaphylaxis appears to be more prevalent in females, although the available literature is dated [63]. It should be noted that fatal and near-fatal anaphylaxis is reported prevalently in males [64,65,66]. Variations in blood levels of female hormones are a recognised, though rare, sex-specific cause of anaphylaxis. Catamenial anaphylaxis is a cyclic presentation at the onset of menstruation, generally presenting in women aged 30–40, but recently described in adolescents as well [67,68]. Progestogen hypersensitivity with systemic or organ-limited manifestations is a similar clinical entity, usually occurring in the second half of the menstrual cycle when progesterone levels are on their peak but also after in vitro fertilisation, with a hypothesised IgE-mediated immediate-type mechanism [69]. Extremely rare cases of anaphylaxis have been described during lactation and breast feeding, possibly due to an imbalance between the effects of oestrogens and progesterone [70]. Oestrogen hypersensitivity due to endogenous or exogenous hormones has been described, but clinical manifestations are typically limited to skin [71]. No increased risk for anaphylaxis has been reported in transgender individuals so far [72]. Dated studies on total serum IgE attribute higher levels to males since birth with a progressive decrease seen in both sexes with age [73,74], but contrasting results are reported on allergen-specific IgE relevant to anaphylaxis. The Swedish BAMSE cohort demonstrated that allergen-specific IgE levels do not significantly differ between sexes independently of age or specific triggers (airborne or food-related). However, sensitization to airborne allergens in particular is significantly associated with males (OR 1.68) [75]. IgE switch in B cells is mediated by IL-4 and IL-13, two cytokines mainly produced by T_H_2 lymphocytes in a process actively influenced by sexual hormones. Both progesterone and oestrogens promote type-2-oriented cells [76]. Basal serum tryptase levels are a useful marker of cellular burden in MC-mediated disorders. Familial alpha-tryptasemia (HaT), a genetic predisposition of autosomal dominant inheritance, is suspected to be associated with an increased risk and severity grade of anaphylaxis, with contrasting evidence [77,78]. Nevertheless, tryptase serum levels are significantly correlated with both sex and age, with higher concentrations in males and in the 18–58 year age range; of note, this significance is lost in the elderly [79]. Despite this, tryptase expression in vitro is upregulated by oestrogens and progesterone in the human MC line HMC1 [80]. HMC1, normal MCs, and the basophil RBL-2H3 line express both progesterone receptors and oestrogen alpha-receptors (Erα). The same cells also express androgen receptors and could be further regulated by testosterone [81]. To date, it is still unknown whether other receptors relevant to anaphylaxis (e.g., Toll-like receptors, Mas-related G-protein coupled receptor member X2—MRGPRX2) are differently expressed on MCs in the two sexes or whether they might be modulated by sexual hormones.

*Recommendations:* Female sex appears to be a predisposing factor for allergic diseases, but available data on a similar relationship with anaphylaxis is unclear. Epidemiologic studies have so far been scarcely informative, as stratification by age, sex, exposure, triggers, and data sources is oftentimes not coherent. Clinicians should be aware of female-predominant forms such as catamenial anaphylaxis, as well as the higher fatality risk observed in males. Future studies should focus on an accurate stratification of afflicted patients in order to improve clinical practice and refine diagnostic and preventive strategies.

## 7. Drug Hypersensitivity

Female sex is a strong risk factor for hypersensitivity reactions (HSRs). Several reviews highlight a higher frequency of self-reported drug allergy in females [82]. For example, data from the European Anaphylaxis Registry shows a higher proportion of drug-induced anaphylaxis than insect- or food-related anaphylaxis in females: among 1825 patients, 1186 were female (65.34%, *p*  <  0.0001) and only 629 were males (34.66%, *p*  <  0.0001). On the other hand, severe anaphylaxis with hospitalisation appears to be more common in men than in women [83]. Sex differences in self-reported drug allergy have not been observed in paediatric populations [84]. The possible mechanisms are not well-understood and are most likely both biological and psychosocial [82]. A recent Italian study using real-world data highlighted that usage of prescription drugs appears to be higher in females, although males are characterised by longer treatment durations [85]. On the other hand, genetic, epigenetic, and hormonal differences between sexes influence both innate and adaptive immune responses, proposed as a possible explanation for female predisposition to drug allergies [86].

The current definition of adverse drug reactions (ADRs) is any unwanted and harmful effect of a drug at a normally tolerated dose; HSRs in turn are categorised by parameters such as timing of reaction or immunopathogenesis. One of the most well-known is the Gell and Coombs classification, where HSRs are divided into four types: type I (immediate IgE-mediated), type II (cytotoxic-, antibody-, and Fc receptor-mediated; cellular), type III (immune complex-mediated), and type IV (delayed-type, T-cell-mediated). Recently, the European Academic of Allergy and Clinical Immunology (EAACI) has published a new nomenclature of allergic diseases and hypersensitivity reactions [87].

The most common drugs related to anaphylaxis are penicillin, sulphonamides, and nonsteroidal anti-inflammatory drugs (NSAIDs) [61]. Immediate HSRs are more frequent in females, often due to an IgE-mediated mechanism as demonstrated by skin tests, both prick and intradermal, and serum drug-specific IgE [82]. It should be noted that although drug provocation tests (DPTs) remain the diagnostic gold standard, a positive challenge does not readily imply an IgE-mediated pathogenetic mechanism. Other mechanisms, such as complement or MRGPRX2 activation, can lead to mast cell and basophil degranulation and mediators’ release. Data is still limited, but HSRs mediated by MRGPRX2 receptors appear to be more frequent in females: radiocontrast media are an example of such a mechanism [88].

Delayed HSRs can be heterogeneous, without a clear association with sex. Usually, delayed HSRs are mediated by T cells and there is a strong correlation with Class I HLA proteins that mediate drug presentation, especially for more severe cutaneous ADRs (SCARs). Research on genetic predisposition is generally focused on race and ethnicity and less on sex [82]; nevertheless, a recent study on 25 patients with allopurinol-induced SCARs demonstrated a prevalence of HLA-B*58:01 expression in females [89].

Allergy drug label is a considerable risk during pregnancy. Penicillin allergy label is the main example, being the first-line treatment in pregnancy-related ailments such as syphilis and group B *Streptococci.* The prevalence of self-reported penicillin allergy in pregnant women is approximately 8%, but just as in the general population, true allergy is confirmed only in 5–10% of cases. The use of second-line antibiotics such as vancomycin, clindamycin, and gentamicin is associated with a higher mortality and a higher frequency rate of ADRs [82,90]. As such, particular care should be taken in penicillin allergy de-labelling, even with mild allergic reactions. However, to date, skin testing followed by an oral challenge in case of negative results is considered a safe procedure for penicillin de-labelling during pregnancy [91].

*Recommendations:* Female sex is associated with a greater incidence of drug hypersensitive reactions, partly due to hormonal and immunological influences. Clinicians should carefully evaluate allergy de-labelling, especially during pregnancy, to reduce their impact and unnecessary use of second-line drugs. Future studies should focus on understanding the underlying molecular mechanisms and signalling pathways involved in drug responses and immune system interactions, informing diagnostic testing and improving therapeutic safety.

## 8. Mastocytosis

Mastocytosis is a rare disease caused by clonal, neoplastic proliferation of abnormal mast cells that accumulate in one or more organ systems. The clinical presentation is heterogenous, ranging from skin-limited disease to more aggressive variants involving bone marrow or multiple organs concurrently; anaphylaxis is often the main presenting symptom. The prevalence of systemic mastocytosis (SM) seems to be slightly greater in women, with a rate of 55–59.4% compared to men [92,93,94]. The prevalence of different mastocytosis subtypes varies between sexes: indolent systemic mastocytosis (ISM) affects mostly women (55–65.2%) [93,95,96,97], while males are mostly affected by advanced forms [95]. Female sex seems to be a protective factor for overall survival and progression-free survival [96,98,99]: in a study by Kluin-Nelemans et al., the overall survival was significantly inferior in males affected by SM, both indolent and advanced (*p* < 0.0001); furthermore, progression-free survival was lower in males (*p* = 0.0002) [100]. Sex appears to be correlated to key clinical differences [95], with cutaneous involvement being more frequent in females independently of subtype [95,96,100]. Non-aggressive forms of SM are strongly related to hypersensitivity reactions presenting as anaphylaxis. Hymenoptera stings are the most common triggers, followed by drugs, food, and others, with the latter being more prevalent in females [92]. Conversely, Hymenoptera venom hypersensitivity reactions are more prevalent in males, an occurrence attributable to different professional activities and hobbies [92,101]. Notably, Hymenoptera venom hypersensitivity strongly correlates with bone marrow mastocytosis, whose prevalence is higher among males [92,95,96,101]. Osteoporosis is a common coexisting condition in SM in both sexes, with a higher incidence in young men and a higher risk of fractures when compared to premenopausal women [95,102,103]. It should be noted that pregnancy may induce exacerbation of mast-cell mediator release symptoms, owing to hormonal influences, although clinical improvement can be noted in approximately one third of expecting women [104,105,106,107,108]. Unexplained osteoporosis or the occurrence of anaphylactic reactions, particularly those induced by Hymenoptera stings or presenting with isolated cardiovascular manifestations, should prompt consideration of systemic mastocytosis. An accurate diagnosis is critical for establishing a comprehensive diagnostic and prognostic framework. This facilitates appropriate therapeutic strategies: a timely initiation of treatment in aggressive forms, prevention of recurrent anaphylaxis in indolent forms (e.g., through lifelong venom immunotherapy), and management of mediator-related symptoms [95,109]. Anaphylaxis during pregnancy and delivery is rare, but mast cell degranulation during labour or caesarean section is possible. Pre-treatment with antihistamines and corticosteroids may be considered, even though evidence supporting premedication in preventing anaphylaxis is poor. Physical triggers and drugs associated with mast cell degranulation should be avoided [110]. Cytoreductive agents are not recommended in pregnancy due to the risk of foetal harm while antihistamines, antileukotrienes, omalizumab, corticosteroids, and epinephrine are considered safe options during pregnancy [107].

*Recommendations:* Sex markedly influences the prevalence and clinical course of mastocytosis. While systemic forms are slightly more frequent in females, males are more affected by aggressive subtypes and are at higher risk for osteoporosis and fractures when compared to premenopausal women; in the same vein, skin involvement and hypersensitivity reactions to drugs and foods are more common in females. Clinicians should focus on prevention and treatment of secondary osteoporosis in men, along with symptom control in women, particularly during pregnancy. The management of mastocytosis in expectant patients should focus on preventing trigger factors for mast cell activation and promoting the use of medications that counteract mast-cell mediator effects.

## 9. Urticaria

Urticaria is a clinical condition characterised by the development of wheals (hives), angioedema, or both. It is classified according to its duration (acute or chronic), and as inducible or spontaneous with relation to identifiable or non-identifiable triggers [111]. Urticaria is a predominantly mast-cell-driven disease due to histamine, platelet-activating factor (PAF), and other cytokines released from activated skin mast cells. Chronic spontaneous urticaria (CSU), defined as the spontaneous appearance of symptoms for more than 6 weeks, affects approximately 1% of the global population and has a significant impact on the quality of life of patients and their families. Epidemiological studies show a higher prevalence among women, with a female-to-male ratio of 2:1 to 4:1 [112]. Gender differences in CSU have been increasingly recognised, influencing not only disease prevalence but also symptomatology and treatment outcomes [113]. Women report more severe symptoms, including higher urticaria activity scores and increased rates of angioedema, contributing to higher healthcare utilisation [113,114]. Fatigue, sleep disturbances, and emotional distress are also more common in female patients, exacerbating disease burden and reducing the quality of life [115]. Furthermore, CSU significantly impacts sexual health in women, particularly those with recurrent angioedema [116]. The mechanisms underlying gender differences in CSU are not fully understood, but hormonal influences appear to play a significant role. Oestrogens may enhance mast cell activation and histamine release, potentially contributing to CSU exacerbations in women [117,118]. Additionally, CSU is frequently associated with autoimmune conditions such as Hashimoto’s thyroiditis, which predominantly affects females, suggesting a possible autoimmune component in gender disparities [119]. Pregnancy presents unique challenges in CSU management. Studies indicate that CSU worsens in 28.9% of pregnant women, likely due to hormonal fluctuations and immune system adaptations during gestation [120]. Additionally, recent research has identified total IgE levels as potential predictors of CSU course during pregnancy, with higher pre-pregnancy IgE levels correlating with more severe disease [121]. Omalizumab has been demonstrated as a safe and effective treatment option during pregnancy, with no reported adverse maternal or foetal effects [122,123]. The impact of menopause on CSU is less well understood. A study found that 23.3% of postmenopausal women with CSU reported symptom onset at menopause, though disease activity remained unchanged in most cases [124]. Treatment strategies for CSU generally follow international guidelines, with antihistamines as first-line therapy and biologics such as omalizumab for refractory cases [111]. Gender-specific factors should be considered when selecting therapeutic options. Omalizumab appears to be safe during pregnancy, while cyclosporine and other immunosuppressants should be used with caution in women of childbearing age [111,122]. Recent evidence suggests potential sex differences in the efficacy of omalizumab, with some studies indicating that female patients may experience a greater clinical benefit compared to males [125]. While in most cases angioedema may coexist with chronic spontaneous urticaria (CSU), it can also occur independently, particularly in forms that are bradykinin-mediated rather than histaminergic. Oestrogen-related angioedema represents a rare subtype of hereditary angioedema with normal C1-inhibitor, predominantly affecting women (87,5%), and is often triggered or worsened by elevated oestrogen levels during puberty, pregnancy, or following the use of oestrogen-containing therapies [126,127]. This form of angioedema is non-pruritic and non-pitting, with the majority of the attacks occurring in laryngeal and abdominal regions [127]. Recurrent attacks are typically unresponsive to antihistamines or corticosteroids, which may lead to diagnostic delays [128]. Oestrogens are known to modulate the kallikrein–kinin system, and their involvement is especially relevant in hereditary angioedema with normal C1-inhibitor (HAE-nC1-INH), particularly in those with F12 gene mutations [126,129]. In this context, recognition of hormonal triggers is essential for appropriate management, which may include withdrawal of oestrogen-based treatments and, in selected cases, the use of bradykinin-targeted therapies [130].

*Recommendations:* The pathogenesis of CSU is complex, involving mast cell activation, autoimmune mechanisms, and inflammatory pathways. Sex and gender differences in CSU are evident in prevalence, symptomatology, and response to hormonal changes. In particular, hormonal fluctuations, pregnancy, and menopause may influence disease activity and should be integrated into management strategies. Understanding these disparities is essential for optimising treatment strategies and improving patient outcomes. Future research should further explore the mechanisms driving these differences to refine gender-specific therapeutic approaches.

## 10. Atopic Dermatitis and Allergic Contact Dermatitis

Atopic dermatitis (AD) is a chronic inflammatory disease characterised by an imbalance of Th2 to Th1 cytokines and alterations in the skin barrier [131]. Available scientific evidence notes a higher incidence in females, owing to a complex interplay between hormonal factors, genetics, skin physiology, lifestyle, and environment, along with sociocultural and psychological factors [132]. The prevalence of AD is moderately higher in males during the first year of childhood and during puberty [133]. This difference is inverted afterwards, with incidence in females being higher than in males, as shown by Japanese, [134], European, and American cohorts [135]. This phenomenon is most evident in asthma, which is another Th2-mediated allergic disease. Extrinsic AD appears correlated to a shift towards a Th2 cytokine pattern, with Th2 lymphocytes being far more prevalent in females after puberty. Androgens induce greater impairment of the skin barrier compared to oestrogens which, on the contrary, appear to be protective; the reverse occurs in the luteal phase in which progesterone induces an opposite effect. In general, and especially in the postmenopausal phase, women are more affected by AD than males, due to a prevalence of the Th2 immunological pattern. In the premenstrual phase, a worsening of AD occurs, also in this case due to the effect of oestrogen and progesterone on Th2 activity whose effects are higher than in the other phases of the cycle; this also occurs during pregnancy, again due to extremely high concentrations of elevated oestradiol and progesterone in these conditions [136]. Intrinsic AD is correlated with skin test positivity to metals, especially nickel, which in turn is more frequent in females. This phenomenon can be explained with more frequent exposure to nickel contained in jewellery, cosmetics, and detergents, which becomes much more significant in women after puberty [137]. Skin objectivity greatly differs between sexes, with localizations to the exposed areas of the head, neck, and hands being more frequent in females.

Key differences have been observed in allergic contact dermatitis (ACD) in terms of prevalence, triggers, and clinical manifestations. Sex-specific differences in skin structure and physiology, lifestyle, occupation, usage of specific personal care products, as well as genetic and hormonal factors play a key role in allergen exposure and responses [138]. Boonchai et al. observed that females exhibit a significantly higher incidence of patch test positivity. While occupation-related haptens are the prevalent triggers in males (e.g., “carba mix”, epoxy resin, n-isopropyl-n-phenyl-4-phenylenediamine), haptens contained in jewellery, personal care, and cleaning products (e.g., colophonium, formaldehyde, neomycin, fragrance mix, nickel) are more common in females. This is further reinforced by localization, with trunk and extremities being more frequent in males and face and hands in females [139].

Available data suggests nervous system involvement in the pathogenesis of ACD via neuropeptide-mediated neurogenic inflammation of skin tissue mediated by components of the immune system, such as mast cells [47,140]. Oestrogens and progesterone can influence the release of neuropeptides, like substance P and calcitonin gene-associated peptide, resulting in different neuroimmune communication pathways [141]. This offers a plausible explanation for the greater vulnerability to ACD in postmenopausal women [139].

Disease severity appears to be significantly greater in females, with a greater impact on quality of life [142]. Studies reporting self-assessments highlight that the perception of morbidity is significantly higher in women than in men. Overall, although the authors found no differences in overall results regarding health-related quality of life, girls were more embarrassed, shy, upset, and sad due to atopic dermatitis. The authors’ findings could influence the educational part of the consultations of children with atopic dermatitis [135]. Environmental factors play a crucial role in explaining key differences between genders in AD. The prevalence of indoor activities versus outdoor activities might influence AD phenotypes: eczema, for example, doubles in prevalence among female children, owing to more frequent indoor ludic activities compared to males [136]. From adolescence onward, exposure to certain stimuli such as cosmetics and cigarette smoke can potentially aggravate gender-specific disease patterns [143]. In addition sex differences could play an important role in microbiota composition, thus inducing increased susceptibility to autoimmune, endocrine, and skin diseases [144].

*Recommendations:* Sex differences in skin physiology, environmental exposures, and neuroinflammatory pathways contribute to distinct dermatitis patterns. Clinicians should advise sex-specific lifestyle adjustments to reduce exposure to triggers, such as tobacco abstinence, regular physical activity, and dietary patterns characterised by an anti-inflammatory effect on the gut microbiota. Research should continue to explore hormonal and environmental determinants to optimise personalised management.

## 11. Gastrointestinal Allergies

The available literature data supports sex as a significant determinant in the incidence of gastrointestinal allergies, specific foods involved, as well as differences in clinical management. During childhood, food allergies have a higher incidence in males, whereas this trend appears reversed during adulthood, with a female-to-male ratio of 1:0.53 [145,146]. Said differences can be explained by multiple underlying factors, chiefly the role of sex hormones. Receptors have been located on lymphocytes, monocytes, eosinophils, basophils, and mast cells, with oestrogens and progesterone inducing Th2 polarisation and potentially favouring the development of allergic diseases [147,148]. Further evidence in the literature points towards an immunosuppressive effect of progesterone, androgens, and glucocorticoids, whereas oestrogens enhance mast cell reactivity, delayed type IV allergic reactions, humoral responses, and autoimmunity. Recent evidence points toward the intestinal microbiota as a determining element in gender differences: environmental and social factors, different dietary patterns, and baseline differences in gut microbiota play a role in immune response and the onset of gastrointestinal allergies [149]. Of note, fruits and berries appear to be more frequent allergy triggers in females (44% versus 24%), whereas the opposite is true for peanuts (43% versus 27%) [28]. The impact on lifestyle appears to be different in both sexes: female patients exhibit a greater degree of dietary and lifestyle adaptation to a diagnosis of food allergies, as well as disease awareness. A recent study noted how males with severe food allergies were less likely to carry an epinephrine auto-injector with them compared to females [150].

*Recommendations:* Food allergy prevalence and triggers vary by sex across the lifespan of an individual, reflecting a complex interplay of hormonal, immunological, and microbiota-related factors. Clinicians should consider these differences in diagnosis and treatment and promote adequate self-management strategies by improving patient awareness. Prospective studies are needed to integrate sex-specific factors into precision medicine approaches.

## 12. Professional Allergies

Occupational allergic diseases are among the most frequent work-related ailments, chiefly affecting the skin and lungs. Workplace-specific risk factors, modalities, and exposure timing to allergens and haptens may vary between males and females. Nevertheless, the definition of gender-related occupational allergy is still debated, although such a differentiation is oftentimes undeniable [151]. According to available epidemiologic data, 16% of late-onset asthma is work related, be it occupational asthma (OA) or work-exacerbated asthma (WEA), and caused by contact with isocyanates, animal dander, moulds, cereals, wood dusts and acrylic monomers, among others [152]. Some gender-related differences in OA have been reported: in a Canadian retrospective study, the prevalence of OA was significantly higher in women, particularly healthcare and retail workers. A possible explanation may lie in increased exposure to moulds, indoor pollutants and cleaning products in females [153]. Occupational rhinitis is considered a risk factor for asthma and is more common in food, textile, and agricultural sectors, with a higher prevalence in females. Notably, the prevalence in males decreases with age [154]. Furthermore, the prevalence of lifetime allergic rhinitis was associated with female gender among US primary farm workers [155], with a subsequent Swedish study confirming these findings [156]. However, prolonged professional exposure to vapour, gases, dusts, and fumes (VGDF) is associated with an increased risk of rhinitis in males. In Northern Europe, the incidence of adult-onset rhinitis was higher in welders, especially in females [157]. The available literature data does not report differences in prevalence among the sexes for hypersensitivity pneumonitis and extrinsic allergic alveolitis [158]. Likewise, the effect of exposure to specific fumes, dusts, and steams on asthma exacerbations can vary between sexes; a German study noted a higher prevalence of asthma caused by flour dust in men, whereas in an American cohort a higher prevalence was found in female hairdressers. Of note are differences in post-exposure phenotypes: the inhalation of inorganic dust causes dyspnoea and bronchospasm in women, whereas organic dust has been shown to cause a reduction in FEV1 in men. Other behavioural factors, such as smoking, seem to have a more pronounced detrimental effect on asthma severity in females [37]. Occupational eczema is a recurrent disease affecting the hands: it can become chronic, with a remarkable impact on quality of life. The most common ailment is contact dermatitis, either allergic or related to irritants, with a prevalence of 90–95% [159,160]. Although genetic predisposition is a risk factor, pathogenesis is mainly related to environmental exposure: forms of so-called “wet work”, characterised by frequent hand washing or prolonged glove usage, are particularly at risk: cleaners, health workers, chefs, fishmongers, cosmetologists, and butchers are such examples. The literature data has noted a prevalence of females in such occupations, leading to higher exposure and more frequent cases of job change [161].

*Recommendations:* Occupational allergy patterns differ by sex, owing to distinct exposure patterns and job roles. Clinicians should promote mandatory and appropriate workplace prevention strategies, including adequate use of protective equipment and, should exposure be unavoidable, specific immunotherapy to treat severe cases. Future studies on workplace-related health effects should consider sex and gender differences in exposure assessment to better guide policy and prevention.

## 13. Conclusions

Within the ever-changing landscape of allergic disease, sex appears to be a crucial element for researchers and clinicians alike to take into consideration when approaching novel treatment strategies.

We summarise our findings below.

### 13.1. Gender as a Key Determinant in Allergy Pathophysiology

The evidence reviewed in this paper confirms that both biological sex and gender-related factors critically influence the development, clinical expression, and treatment response of allergic diseases. Hormonal influences, immune modulation, and genetic predispositions interact to create distinct disease phenotypes between male and female patients (Figure 1 and Figure 2). Clinicians should integrate these insights into patient assessment and therapeutic considerations, tailoring their interventions to sex- and gender-specific disease patterns.

### 13.2. The Need for a Gender-Specific Approach in Research

Historically, allergic disease research has not sufficiently differentiated between male and female subjects. Our analysis underlines the urgent requirement for clinical trials and epidemiological studies that stratify outcomes based on sex and gender. Future research should take these stratifications into consideration to uncover the mechanisms underlying these differences and to guide the development of targeted treatments.

### 13.3. Implications for Clinical Practice and Personalised Therapy

The interplay between gender and immune function suggests that a “one size fits all” approach is suboptimal for managing allergic conditions. Incorporating gender-specific diagnostic criteria and treatment considerations could result in improved management outcomes, including more targeted therapeutic interventions that consider hormonal status, immune response variability, and metabolic differences. Clinicians are encouraged to integrate sex- and gender-informed strategies into daily practices, aiming to enhance patient outcomes.

### 13.4. Future Research Directions and Policy Recommendations

To advance this field, future research should focus on elucidating the molecular pathways that drive gender differences in allergic responses, as well as identifying robust biomarkers for diagnosis and treatment monitoring. Furthermore, interdisciplinary collaboration—merging insights from immunology, endocrinology, and social science—is essential to establish comprehensive guidelines that reflect both biological and societal influences on health. Founding bodies and policy makers should support an integrated approach to ensure research translates into clinical improvements.

### 13.5. Towards Precision Medicine in Allergy and Immunology

Embracing a gender-informed paradigm represents a decisive shift towards precision medicine. By tailoring interventions to the unique characteristics of each patient, clinicians can achieve enhanced disease control and quality of life. This position paper advocates for integrating gender analysis into both research protocols and routine clinical care to ensure that treatment strategies are as effective and equitable as possible.

These conclusions synthesise the current state of evidence while setting a clear agenda for ongoing research and clinical innovation. By explicitly linking findings to clinical practice and research priorities, we aim to invite stakeholders to reconsider traditional methodologies in favour of more nuanced sex- and gender-aware approaches that can potentially transform patient care. Additionally, exploring environmental and lifestyle factors, paediatric versus adult differences, as well as emerging biotechnologies (e.g., omics approaches) may further refine our understanding of these complex interactions.

## Figures and Tables

**Figure 1 ijms-26-09605-f001:**
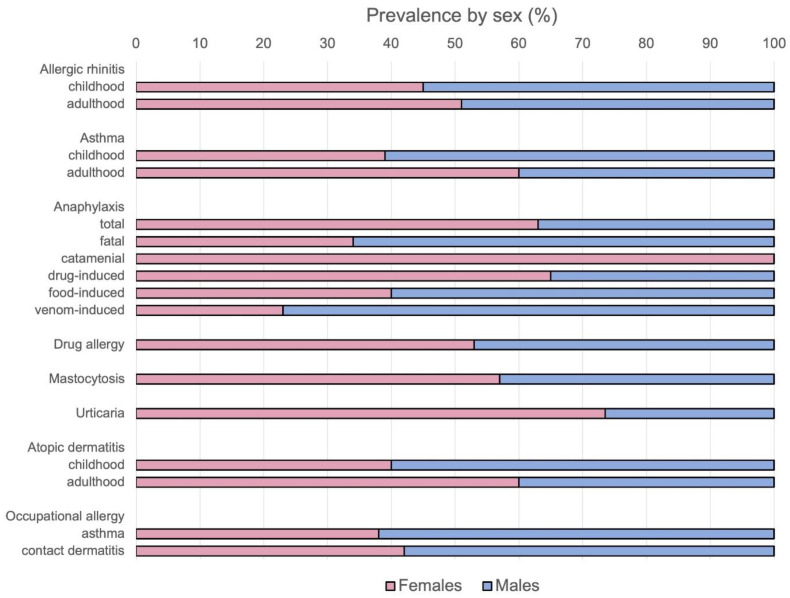
Sex-specific prevalence of the investigated allergic conditions, further subdivided by age group (childhood vs. adulthood) and disease subtype. The graph highlights prevalence pattern shifts between males and females, as well as the joint modulating effects of age and specific disease patterns. (Image generated with BioRender).

**Figure 2 ijms-26-09605-f002:**
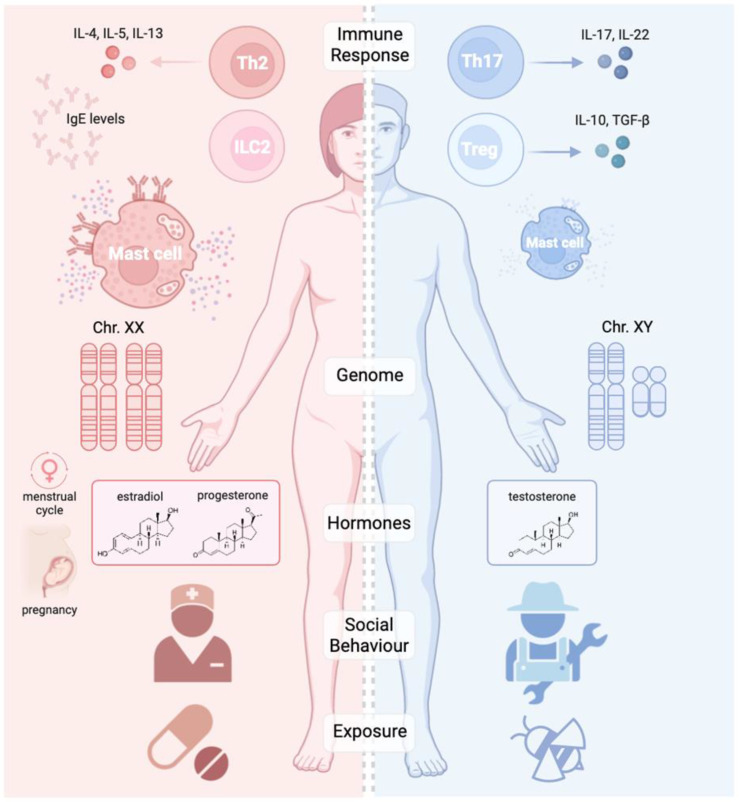
Schematic representation of key differences in biological pathways and prevalent triggers modulating allergic diseases in males vs. females. These mechanisms explain observed clinical differences between male and female patients across the conditions investigated. (Image generated with BioRender).

## Data Availability

No new data were created or analysed in this study. Data sharing is not applicable to this article.

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
