# Peer review of "Gender and Allergy: Mechanisms, Clinical Phenotypes, and Therapeutic Response—A Position Paper from the Società Italiana di Allergologia, Asma ed Immunologia Clinica (SIAAIC)"

_ijms, 2025, doi:10.3390/ijms26199605_

Round 1

Reviewer 1 Report

Comments and Suggestions for Authors

General Comment

The position paper by Ventura, et al addresses the role of sex/gender in allergy, covering mechanisms, clinical phenotypes, and therapeutic implications. The manuscript is comprehensive, well-referenced, and clinically relevant. It integrates immunological, epidemiological, and clinical evidence, providing valuable recommendations. However, several issues warrant improvement before publication. The manuscript is, in general, too verbose. The removal of redundancies is strongly advised.

Major comments

  1. Sex vs. Gender (Multiple Sections, e.g., Lines 66–80, 243–305, 367–409) - While the introduction acknowledges the differences, sex and gender are used interchangeably throughout the manuscript. Please revise noting that sex should denote biological differences, while gender refers to sociocultural constructs. 

  2.  line 67-68: “trials were based on the prevailing notion that male and female patients exhibited near-identical molecular and phenotypical characteristics” this sentence is too categorical: early studies often excluded women.

  3. Methods: Please clarify which search terms and time frames were used in PubMed/Medline? Were inclusion/exclusion criteria applied? How was the consensus process structured (e.g., Delphi)?

  4. The methods subsections should be moved into a new section at the discretion of the authors (i.e., Immunological aspects of allergic disease should be the first section of the discussion, not methods)
  5. Lines 108-109:  The statement “oestrogens are considered enhancers of the immune response, while androgens and progesterone act as immunosuppressants” is too broad. Please provide more nuance (e.g., ERα vs ERβ, progesterone pro-inflammatory in ILC2 responses).

  6. Lines 116-119: the same happens here. I assume the authors mean T1/T2 and not Th1/Th2 responses. Moreover, the prevailing response varies according to the stage of the pregnancy (i.e., T2 responses are important during implantation and 1st trimester). Please revise.
  7. Lines 171-172: do the authors mean that AR symptoms worsen during pregnancy or that rhinitis worsens during pregnancy potentially due to rhinitis gravidarum?
  8. lines 182-183- which gender-specific treatment strategies would the authors recommend? How would the authors recommend an approach accounting for hormonal fluctuations and how do the authors see this recommendation being seen by the general public?
  9. Lines 221-223190: Is the claim “testosterone reduces Th2-driven inflammation and modulates airway remodeling” based on human studies or just animal studies? Also, Please correct to T2-driven

  10. Anaphylaxis section is often contradictory, too verbose, and warrants clarity. There is hardly enough evidence to categorically state that anaphylaxis is more frequent in females vs males. Most studies are heavily biased. The same happens for severity. Studies are all-over the place. In fact, fatal anaphylaxis is more prevalent in males, and it is less likely to have biases for these data. Also, what credible evidence is there for progestogen hypersensitivity?! Please revise the expression of hormone receptors by mast cells. Moreover, is HaT a disorder? Is there definitive evidence showing that HaT increases the risk and severity grade of anaphylaxis, or is this still a matter of debate?

  11. Drug allergies: please change to the more correct Drug hypersensivity. Moreover, please revise the structure, it is too condensed into a single paragraph.
  12. Lines 325-326: "Therefore, female sex, especially during repro- 325
    ductive age, is a risk factor for autoimmune diseases and atopic conditions." why is this sentence here?
  13. In the anaphylaxis section, there is no evidence of gender-differences for MRGPRX2 activation, but for drug hypersensitivity there is?
  14. Mastocytosis section: “female sex seems to be a protective factor for overall survival and progression-free survival” there is no evidence showing this in such a categorical way. Please tone down. Also, most studies do not show a significant difference between the prevalence of mastocytosis in males vs females. In fact, it seems to be more frequent in boys than in girls. The sentence about worsening of symptoms could be used for asthma, or rhinitis... Please revise. Please provide references for the indication and protocols for premedication. Oral cromolyn is used for chronic treatment, not as premedication.

  15. Conclusions (Lines 590–633): Well-structured but somewhat generic. The conclusions should explicitly link back to clinical practice and research priorities

  16. References: Some references are recent and appropriate, but a few are tangential (e.g., ref. 3 on access to healthcare in LGBTQ+ populations, l.661–662). While relevant to gender medicine broadly, they may appear out of scope. Others should be cross-checked for accuracy and correct formatting (e.g., missing page ranges, inconsistent journal abbreviations).

Minor Issues

  1. The names of the authors seem to be reversed (Last name, first name).
  2. Language and Style: please revise the whole text carefully for missing and repetitive words (e.g., line 38: “characteristics of the the most common allergic diseases”)

  3. Figures: The legends are too brief (“Image generated with BioRender”). Provide explanatory captions clarifying what is depicted and how it integrates into the text.

  4. Consistency: “oestrogen,” and “estrogen.” used interchangeably. Choose either UK or US English consistently.

  5. Recommendations: Each subsection ends with “Recommendations,” but the style and depth vary. Some are concise (e.g., AR), others generic (e.g., anaphylaxis).

Author Response

To the Editorial Board of the International Journal of Molecular Sciences,
and
to Dr. Stefania Nicola – Guest Editor.

Dear Editor(s),

On behalf of my colleagues and co-authors I would like to thank the Reviewers for their time and invaluable insights, according to which we have vigorously revised the paper’s form and contents. Therefore, we would like to offer a point-to-point reply to each Reviewer’s observations.

Reviewer 1

“The manuscript is, in general, too verbose. The removal of redundancies is strongly advised.” - We have thoroughly reviewed the text to optimize its contents and improve readability.

Major comments

  1. Sex vs. Gender (Multiple Sections, e.g., Lines 66–80, 243–305, 367–409) - While the introduction acknowledges the differences, sex and gender are used interchangeably throughout the manuscript. Please revise noting that sex should denote biological differences, while gender refers to sociocultural constructs.  - We have amended the manuscript for uniformity, as well as appropriate usage of "sex” and “gender”.
  2.  line 67-68: “trials were based on the prevailing notion that male and female patients exhibited near-identical molecular and phenotypical characteristics” this sentence is too categorical: early studies often excluded women.” – we have amended the passage according to this suggestion.
  3. “Methods: Please clarify which search terms and time frames were used in PubMed/Medline? Were inclusion/exclusion criteria applied? How was the consensus process structured (e.g., Delphi)?” - Literature search was conducted on MEDLINE and PubMed, combining search terms such as “sex differences”, “gender differences” and “allergy” with the specific diseases being discussed. There was no restriction on publication year, and we focused on English-language papers. Titles and abstracts were screened for relevance, while non-human studies and case reports were excluded. Each section was drafted by the designated Authors, circulated among all co-Authors for collective review, discussion, and integration, with disagreements resolved through group discussion until consensus was reached. Selection of the nine diseases discussed, as well as consensus throughout the entire process, was achieved using the Delphi methodology. The paper has been amended to include this information.
  4. “The methods subsections should be moved into a new section at the discretion of the authors (i.e., Immunological aspects of allergic disease should be the first section of the discussion, not methods)” – we concur with the Reviewer, the current section numbering is unintended: we have edited the document to restore proper numbering.
  5. Lines 108-109:  The statement “oestrogens are considered enhancers of the immune response, while androgens and progesterone act as immunosuppressants” is too broad. Please provide more nuance (e.g., ERα vs ERβ, progesterone pro-inflammatory in ILC2 responses).” – the passage has been edited according to the Reviewer’s suggestions.
  6. Lines 116-119: the same happens here. I assume the authors mean T1/T2 and not Th1/Th2 responses. Moreover, the prevailing response varies according to the stage of the pregnancy (i.e., T2 responses are important during implantation and 1st trimester). Please revise.” – the passage has been edited according to the Reviewer’s suggestions.
  7. “Lines 171-172: do the authors mean that AR symptoms worsen during pregnancy or that rhinitis worsens during pregnancy potentially due to rhinitis gravidarum?” – both worsening of allergic rhinitis and onset of rhinitis gravidarum are implicated in worsening symptoms during pregnancy; the passage has been overhauled for clarity.
  8. “Lines 182-183- which gender-specific treatment strategies would the authors recommend? How would the authors recommend an approach accounting for hormonal fluctuations and how do the authors see this recommendation being seen by the general public?” - the key strategy is early diagnosis of AR, in order to achieve the best possible management according to sex/gender and age differences: the relevant passage has been edited to improve clarity on this matter.
  9. Lines 221-223: Is the claim “testosterone reduces Th2-driven inflammation and modulates airway remodeling” based on human studies or just animal studies? Also, Please correct to T2-driven.” - we thank the reviewer for the suggestions. The sentence “Th2-driven” has now been changed as suggested. Concerning the first question, according to the reference we cited and its linked references, the sentence is true also in human studies.
  10. “Anaphylaxis section is often contradictory, too verbose, and warrants clarity. There is hardly enough evidence to categorically state that anaphylaxis is more frequent in females vs males. Most studies are heavily biased. The same happens for severity. Studies are all-over the place. In fact, fatal anaphylaxis is more prevalent in males, and it is less likely to have biases for these data. Also, what credible evidence is there for progestogen hypersensitivity?! Please revise the expression of hormone receptors by mast cells. Moreover, is HaT a disorder? Is there definitive evidence showing that HaT increases the risk and severity grade of anaphylaxis or is this still a matter of debate?” - we have extensively reworked this section to remedy the points of concern raised by the Reviewer:  the paragraph has been shortened and some redundant sentences on cellular/molecular mechanisms have been removed. Regarding available evidence on sex-specific differences in frequency and severity of anaphylaxis, we concur that literature data is unclear, as we stated in the “recommendations” section: the studies from UK and Korea/Taiwan, reported in the text among many others which we excluded for brevity, were aimed to be examples of the inhomogeneous results on this topic. In agreement with the Reviewer’s observations, the biases of the cited population studies have been clarified in the main section. Along with epidemiology, we also examined ex vivo and in vitro data supporting the uncertainty of sex-related differences in anaphylaxis, namely IgE (higher levels in males but antibody production positively regulated by female hormones favouring Th2/T2 differentiation) and tryptase (higher levels in males, although its production in mast cells is upregulated by female hormones). The increased fatality rates in males versus females has consistently been reported for drugs and idiopathic anaphylaxis.
    Progesterone hypersensitivity is a rare but intriguing condition first described by Metcalfe’s group in 1998 and published in the New England Journal of Medicine. Since then, several cases have been reported, purportedly triggered by endogenous or exogenous hormone (for example, nine such cases have been reviewed in Annals of Allergy, Asthma & Immunology 2003, 90:469-477, eight patients in JACIP 2016;4:723-9, one case in JACIP 2017, 5:852 –854). Hypersensitivity was documented by positive skin tests; successful desensitization or response to omalizumab quite convincingly suggesting an IgE-mediated mechanism.
    Acting on the Reviewer’s suggestion, we edited the definition of HaT as a genetic predisposition of autosomal dominant inheritance. On HaT’s role as a risk factor for anaphylaxis and its severity, literature evidence is contrasting: a negative conclusion comes from the comprehensive revision published in Eur Ann Allergy Clin Immunol (Vol 55, N.4, 152-160, 2023); alternatively, a recent paper in JACIP (J Allergy Clin Immunol Pract. 2025 Jun;13(6):1449-1456.e4) correlates severity with the number of TPSAB1 copies. As suggested by the Reviewer, the passage has been revised and the two references presenting these opposing results have been added. The passage on expression of hormone receptors by mast cells has been revised.
  11. Drug allergies: please change to the more correct Drug hypersensivity. Moreover, please revise the structure, it is too condensed into a single paragraph.” – the section has been reworked as per the Reviewer’s suggestions.
  12. Lines 325-326: "Therefore, female sex, especially during repro- 325
    ductive age, is a risk factor for autoimmune diseases and atopic conditions." why is this sentence here?
    ” – this specific passage was intended to highlight the possible role of genetic, epigenetic and hormonal differences in explaining female propensity for drug allergy. We have reworked it for clarity.
  13. In the anaphylaxis section, there is no evidence of gender-differences for MRGPRX2 activation, but for drug hypersensitivity there is?” - an increased expression of MRGPRX2 on female mast-cells has not been described so far. On the other hand, Dinah Foer et al (J Allergy Clin Immunol Pract 2023;11:492-99) examined electronic health record data of an adult cohort with a MRGPRX2 -activating drug exposure, highlighting how female sex was associated with increased odds of reaction. It should be noted that the discrepancy between these two  observations may depend on several factors, such as variations in responsiveness, possible genetic variants, post-transcriptional modifications, different binding capacity in addition to in vivo involvement of other target cells not expressing the receptor during the course of the adverse reaction (see for example Curr Allergy Asthma Rep 25, 5, 2025).
  14. Mastocytosis section: “female sex seems to be a protective factor for overall survival and progression-free survival” there is no evidence showing this in such a categorical way. Please tone down. Also, most studies do not show a significant difference between the prevalence of mastocytosis in males vs females. In fact, it seems to be more frequent in boys than in girls. The sentence about worsening of symptoms could be used for asthma, or rhinitis... Please revise. Please provide references for the indication and protocols for premedication. Oral cromolyn is used for chronic treatment, not as premedication.” – we would like to thank the Reviewer for this comment, raising several important issues upon which we have improved this section. More specifically:
    1. “female sex seems to be a protective factor for overall survival and progression-free survival” there is no evidence showing this in such a categorical way. Please tone down”
      We have reworked the section accordingly and added supplemental references to clarify our statements. A higher risk of both progression and death, worse performance status, and organomegaly was significantly associated with males in a work by Trizuljak et al. (Allergy Eur J Allergy Clin Immunol 2020;75(8):1923–34) Likewise, the work of Kluin-Nelemans HC et al highlighted how overall survival was significantly inferior in males, and also within the WHO sub-categories indolent SM, aggressive SM (ASM) and SM-AHN; male sex has a major impact on clinical features, disease progression, and survival in mastocytosis. Male patients have an inferior survival, which seems related to the fact that they more frequently develop a multi-mutated AdvSM associated with a high-risk molecular background.
    2. “most studies do not show a significant difference between the prevalence of mastocytosis in males vs females. In fact, it seems to be more frequent in boys than in girls”
      Mastocytosis is a disease characterized by many variants. In their paper “Prevalence of hypersensitivity reactions in various forms of mastocytosis: A pilot study of 2485 adult patients with mastocytosis collected in the ECNM registry” published in Allergy last year, Niedoszytko et al reported on 2485 adult patients with diagnosis of mastocytosis, collected in the European registry between 2012 and 2019 in 27 centres: among them, 55.5% were women. Similarly, a Swedish study by Bergstrom et al. found that 59.4% out of 2040 adults with mastocytosis were females. According to a study by Cohen et al. on a Danish population afflicted by mastocytosis, 59.9% of the cohort were females. When considering cutaneous mastocytosis, both smouldering and indolent systemic forms, women were the most affected (65.9%, 55.2%, 50.8% respectively). The data mentioned refers to the overall prevalence of mastocytosis, all variants considered. When analysed by subtype, differences emerge between sexes, which we summarized for conciseness. Following the Reviewer’s advice, we have reworked the section and its bibliographic references.
    3. “the worsening of symptoms could be used for asthma, or rhinitis... Please revise”.
      When referring to symptoms of mastocytosis, we discuss those that resulting from mast-cells mediator release such as flushing, pruritus, hypotension, anaphylaxis, abdominal pain, diarrhoea etc. Pregnancy can be complicated by systemic mastocytosis, with worsening of the manifestation described above. As described in a review by Lei et al. (“Management of mastocytosis in pregnancy: a review”. JACI in Practice), mast-cells mediators release can be facilitated by physical and psychological stress, infections, medications, etc.
      Other authors already used the terms “worsening” and “improvement” when referring to mast-cells mediators’ symptoms. For example, in 2011, the Spanish Network of Mastocytosis (REMA) studied the clinical impact of mastocytosis in pregnancy, with Matito et al. concluding that symptoms may improve or exacerbate during pregnancy and that “mastocytosis shows a heterogeneous clinical behaviour during pregnancy, as happens in other diseases where MC play a relevant role (e.g. asthma)”. In their paper “Mastocytosis in pregnancy” (published in Immunol Allergy Cin N Am in 2023), Arora et al denote similarities with allergic diseases and state: “up to one-third of pregnant patients can experience improvement in symptoms during pregnancy, similar to other autoimmune diseases such as systemic lupus erythematosus and also allergic diseases such as asthma”. In the revised version of the paper, we have slightly changed the sentence with “exacerbation of mast cell mediator release symptoms” instead of “worsening of mast cell mediated symptoms”.
  1. “Please provide references for the indication and protocols for premedication. Oral cromolyn is used for chronic treatment, not as premedication.”
    Acting on the Reviewer’s observation, we have reworked this subsection, removing references to oral cromolyn sodium as possible premedication and added that evidence regarding premedication in preventing anaphylaxis is poor. There is one paper, based on expert opinion, that suggests the usage of steroids, antihistamines and potentially oral cromolyn sodium as premedication of labour and delivery. However, it must be noted that studies have not shown efficacy of such treatment in recurrent anaphylaxis. (Watson KD, Arendt KW, Watson WJ, et al. Systemic mastocytosis complicating pregnancy. Obstet Gynecol 2012;119:486–9).

  1. Conclusions (Lines 590–633): Well-structured but somewhat generic. The conclusions should explicitly link back to clinical practice and research priorities” – we would like to thank the Reviewer for this suggestion, upon which we have reworked the conclusions’ statements to provide a clear, actionable recommendation for clinicians and researchers.
  2. References: Some references are recent and appropriate, but a few are tangential (e.g., ref. 3 on access to healthcare in LGBTQ+ populations, l.661–662). While relevant to gender medicine broadly, they may appear out of scope. Others should be cross-checked for accuracy and correct formatting (e.g., missing page ranges, inconsistent journal abbreviations).” – we have carefully reviewed and revised the paper’s reference list, ensuring accuracy and adherence to the American Medical Association standard (11th edition). We would like to clarify that some of the references cited, while appearing tangential (e.g. Hickey PM et al., ref 3), were intentionally included to emphasize that gender medicine plays a critical role in shaping healthcare needs, encompassing pathogenic mechanisms, disease phenotypes and treatment strategies. Our paper focuses primarily on allergic diseases, but one of its key messages is that management cannot be disentangled from broader sex- and gender-related considerations. With the inclusion of these references, we aim to underscore how profoundly healthcare delivery and outcomes can differ between sexes and even more so across different gender identities. We therefore considered these references useful to place our work within the wider field of gender medicine.

Minor Issues

  1. “The names of the authors seem to be reversed (Last name, first name).” – the Authors’ names have been edited accordingly.
  2. Language and Style: please revise the whole text carefully for missing and repetitive words (e.g., line 38: “characteristics of the the most common allergic diseases”)” – we have thoroughly reviewed the manuscript.
  3. Figures: The legends are too brief (“Image generated with BioRender”). Provide explanatory captions clarifying what is depicted and how it integrates into the text.” – thank you for your suggestion. We have reworked the captions for both figures.
  4. “Consistency: “oestrogen,” and “estrogen.” used interchangeably. Choose either UK or US English consistently.” – the paper has been written in UK English. We have revised the text to ensure consistency.
  5. Recommendations: Each subsection ends with “Recommendations,” but the style and depth vary. Some are concise (e.g., AR), others generic (e.g., anaphylaxis).” – we thank the Reviewer for the suggestion: each recommendation section has been overhauled to improve content and tone consistency.

Reviewer 2 Report

Comments and Suggestions for Authors

The authors prepared this position paper, commissioned by SIAAIC, on a highly relevant and timely topic such as Gender Medicine, applied in this case to Allergology.
The main allergic diseases are reviewed from a gender perspective, encompassing biological, immunoendocrinological, genetic and epigenetic, environmental, and psychosocial aspects, following a well-structured and accessible subdivision into paragraphs. Each section is accompanied by specific recommendations, as well as by two clear and well-designed figures.

The strength of this work lies in its ability to integrate epidemiological and clinical-therapeutic evidence with a solid cellular and molecular basis from a translational perspective. This is consistently distributed across the different sections, enriching the quality of the manuscript without overburdening it, while remaining within the current limits of gender-related allergology and the availability and coherence of existing data. This approach is also well aligned with the scope of a journal such as IJMS and ultimately leads to a perspective of Precision Medicine, with methodological insights for both clinical practice and research.

The conclusions are consistent with the data collected, highlighted, and discussed.
The bibliography is very extensive and includes very recent references.

Some remarks and suggestions:

  • In the paragraph on asthma, it could be useful to specifically refer to neutrophilic inflammation in obese adult women with asthma and/or to the potential gender-specific role of mTOR inhibition, as an example of gender implications in asthma pheno-endotyping and gender pharmacology (see, for example, Zhang P., Zein J. Novel Insights on Sex-Related Differences in Asthma. Curr. Allergy Asthma Rep. 2019;19:44. doi: 10.1007/s11882-019-0878-y. De Martinis M et al. Sex and Gender Aspects for Patient Stratification in Allergy Prevention and Treatment. Int J Mol Sci. 2020 Feb 24;21(4):1535. doi: 10.3390/ijms21041535. PMID: 32102344; PMCID: PMC7073150).
  • Line 480: “this phenomenon…disease”: this sentence is redundant in the context of this paragraph and I suggest removing it.
  • Regarding the paragraph on dermatitis (atopic dermatitis), I suggest systematically mentioning both atopic dermatitis and allergic contact dermatitis, with their sex- and age-related characteristics. Furthermore, the contribution of neurogenic inflammation may also show sexual dimorphism (see, for example, Boonchai W. et al., Gender differences in allergic contact dermatitis to common allergens. Contact Dermat. 2024;90:458–465. doi: 10.1111/cod.14479; Cetinkaya A. et al., Effects of estrogen and progesterone on the neurogenic inflammatory neuropeptides: Implications for gender differences in migraine. Exp. Brain Res. 2020;238:2625–2639. doi: 10.1007/s00221-020-05923-7; Aitella E. et al., Neurogenic Inflammation in Allergic Contact Dermatitis. Biomedicines. 2025 Mar 7;13(3):656. doi: 10.3390/biomedicines13030656. PMID: 40149632; PMCID: PMC11940366).
  • Regarding the paragraph on drug allergy, the reference to the new EAACI nomenclature of allergic diseases and hypersensitivity reactions is highly relevant.
  • Line 491: I recommend replacing skin prick test with skin tests or similar.
  • Line 570: harmonize the font of although with the rest of the text.
  • I recommend adding a short explanatory caption for both figures.
  • The text is overall well written in clear and well-structured English.
    Please check for consistency between oestrogen vs estrogen.

  • Line 168: Gender differences may influence response to treatment, with a recent Chinese study suggesting better short-term efficacy of subcutaneous immunotherapy (SCIT) on clinical outcomes, although the efficacy tends to converge between sexes over longer times: please clarify the implication and specify which gender shows better response.

The work is therefore worthy of publication.

Author Response

To the Editorial Board of the International Journal of Molecular Sciences,
and to Dr. Stefania Nicola – Guest Editor.

Dear Editor(s),

On behalf of my colleagues and co-authors I would like to thank the Reviewers for their time and invaluable insights, according to which we have vigorously revised the paper’s form and contents. Therefore, we would like to offer a point-to-point reply to each Reviewer’s observations.

Reviewer 2

Some remarks and suggestions:

  • In the paragraph on asthma, it could be useful to specifically refer to neutrophilic inflammation in obese adult women with asthma and/or to the potential gender-specific role of mTOR inhibition, as an example of gender implications in asthma pheno-endotyping and gender pharmacology (see, for example, Zhang P., Zein J. Novel Insights on Sex-Related Differences in Asthma. Curr. Allergy Asthma Rep. 2019;19:44. doi: 10.1007/s11882-019-0878-y. De Martinis M et al. Sex and Gender Aspects for Patient Stratification in Allergy Prevention and Treatment. Int J Mol Sci. 2020 Feb 24;21(4):1535. doi: 10.3390/ijms21041535. PMID: 32102344; PMCID: PMC7073150).” - we would like to thank the Reviewer for this remark. In the revised version we added a sentence dedicated to what he said and included the references he suggested.
  • Line 480: “this phenomenon…disease”: this sentence is redundant in the context of this paragraph and I suggest removing it.” – the sentence has been removed, as suggested.
  • Regarding the paragraph on dermatitis (atopic dermatitis), I suggest systematically mentioning both atopic dermatitis and allergic contact dermatitis, with their sex- and age-related characteristics. Furthermore, the contribution of neurogenic inflammation may also show sexual dimorphism (see, for example, Boonchai W. et al., Gender differences in allergic contact dermatitis to common allergens. Contact Dermat. 2024;90:458–465. doi: 10.1111/cod.14479; Cetinkaya A. et al., Effects of estrogen and progesterone on the neurogenic inflammatory neuropeptides: Implications for gender differences in migraine. Exp. Brain Res. 2020;238:2625–2639. doi: 10.1007/s00221-020-05923-7; Aitella E. et al., Neurogenic Inflammation in Allergic Contact Dermatitis. Biomedicines. 2025 Mar 7;13(3):656. doi: 10.3390/biomedicines13030656. PMID: 40149632; PMCID: PMC11940366).” – thank you for your invaluable suggestions, upon which we have revised and improved this section with the inclusion of allergic contact dermatitis. Appropriate references have been selected and added to the manuscript.
  • Line 491: I recommend replacing skin prick test with skin tests or similar.” – the text has been amended.
  • Line 570: harmonize the font of although with the rest of the text.” – the mistake has been corrected.
  • I recommend adding a short explanatory caption for both figures.” – thank you for your suggestion. We have reworked the captions for both figures.
  • The text is overall well written in clear and well-structured English. Please check for consistency between oestrogen vs estrogen.” – the paper has been thoroughly revised to ensure consistency with UK English.
  • Line 168: Gender differences may influence response to treatment, with a recent Chinese study suggesting better short-term efficacy of subcutaneous immunotherapy (SCIT) on clinical outcomes, although the efficacy tends to converge between sexes over longer times: please clarify the implication and specify which gender shows better response.” - unfortunately, in the original revision we did not specify in which gender SCIT has the best short-term efficacy; we clarified in the revised document that this occurs in females, while males are less adherent due to poorer early results, although the efficacy tends to converge between sexes over longer times.